# VIDEO-SALMONN 2: CAPTION-ENHANCED AUDIO-VISUAL LARGE LANGUAGE MODELS

## ABSTRACT

We present video-SALMONN 2, a family of audio-visual large language models that set new state-of-the-art (SOTA) results in video description and question answering (QA). Our core contribution is multi-round direct preference optimisation (MrDPO), paired with a caption-quality objective that jointly rewards completeness and factual accuracy. Unlike standard DPO with a fixed reference policy, MrDPO periodically refreshes the reference by bootstrapping from a newly re-initialised lightweight adapter trained on the latest preferences, avoiding reference staleness and enabling continual improvement. This strategy produces captions that are consistently more detailed and accurate than those from proprietary systems such as GPT-4o and Gemini-1.5 Pro. We further distil these gains by using our model to generate a high-quality video-caption corpus for supervised fine-tuning of new models, transferring benefits beyond captioning to strong performance on complex video-QA tasks. Across widely used audio-visual and visual-only understanding benchmarks (including Video-MME, WorldSense, AVUT, Video-Holmes, Daily-Omni, MLVU, and LVBench), our 3B and 7B models achieve SOTA results at comparable scales, while the 72B model surpasses all other open-source systems. Our source code, models, and data will be released.

## 1 INTRODUCTION

The remarkable success of large language lodels (LLMs) in text-based tasks, where they have approached human-level performance, has inspired widespread efforts to extend their capabilities to multimodal understanding (OpenAI et al., 2024; Dubey et al., 2024; Yang et al., 2025a). A prevalent strategy is to train modality adapters/aligners that bridge pretrained multimodal encoders and a textual LLM, allowing the LLM's knowledge to interpret non-text inputs and produce meaningful outputs. Over the past two years, numerous multimodal LLMs following this paradigm have appeared across modalities, including image and silent video (Liu et al., 2024b;a; Li et al., 2023; Bai et al., 2023; Lin et al., 2023; Chen et al., 2023; Lin et al., 2024; Chen et al., 2024; Li et al., 2024; Zhang et al., 2024b; Bai et al., 2025; Li et al., 2025; Zhang et al., 2025), audio (Wu et al., 2023; Tang et al., 2024b; Chu et al., 2023; 2024; Gong et al., 2024; 2023; Tang et al., 2024c; Zheng et al., 2024), and audio–visual understanding (Team et al., 2024; Cheng et al., 2024; Sun et al., 2024; Fu et al., 2024; 2025b; Tang et al., 2024d; Sun et al., 2025).

High-quality text descriptions paired with data are fundamental to training effective multimodal LLMs. Contemporary systems often use captioning as a core objective in pre-training or supervised fine-tuning (SFT), aligning multimodal encoder outputs with a textual LLM's input space and thereby enabling event recognition and reasoning. Thus, collecting detailed, faithful, low-hallucination captions that are tightly aligned with the inputs is critical for robust multimodal understanding. Yet video captioning remains difficult: videos couple rich intra-frame spatial content with audio–visual events evolving over time. Progress has been limited by unreliable quantitative metrics and limited training strategies for caption quality and by the common practice of discarding the audio stream despite its complementary value. These gaps not only impair caption generation but also constrain downstream performance on general tasks such as video question answering (QA).

We introduce video-SALMONN 2, a family of multimodal LLMs that ingest both audio and visual inputs with a primary focus on detailed, holistic audio–visual captioning. Building on a pretrained visual LLM, video-SALMONN 2 is trained on audio-only corpora and on videos with paired audio tracks, enabling the model to see and hear simultaneously and to capture fine-grained temporal

interactions across modalities. To accurately assess and optimise caption quality, we propose new captioning metrics that evaluate completeness and factuality and use them as reward signals for reinforcement learning (RL) via direct preference optimisation (DPO) (Rafailov et al., 2024). We further introduce multi-round DPO (MrDPO), which iteratively updates the reference policy using a lightweight adapter, driving continual improvements in captioning. Finally, we use the MrDPO-trained model to generate higher-quality captions and perform SFT on this improved data to train new models. Experiments show that our final models retain state-of-the-art (SOTA) captioning performance and transfer gains to general video understanding. On widely used audio–visual and visual-only video QA benchmarks, including Video-MME, WorldSense, AVUT, Video-Holmes, DailyOmni, MLVU, and LVBench, our final models with 3 billion (B) and 7B parameters achieve SOTA results at comparable scales, and our 72B model outperforms proprietary systems such as GPT-4o and Gemini-1.5 Pro. Our main contributions can be summarised as follows:

• We develop video-SALMONN 2, a family of strong audio–visual LLMs with enhanced video-captioning ability that surpasses proprietary systems such as GPT-4o and Gemini-1.5 Pro.
• We introduce a lightweight evaluation pipeline that uses text-only LLMs to estimate missing and hallucinated audio–visual events by decomposing caption assessment into LLM-friendly sub-steps, enabling metric-driven RL for caption optimisation. We also release a human-annotated video-captioning benchmark to validate our evaluation approach.
• We propose MrDPO, a caption-optimisation method that periodically updates the DPO reference policy by merging and re-initialising a low-rank adaptation (LoRA) (Hu et al., 2022) proxy, and stabilises training by smoothing the loss with SFT on ground-truth captions.
• We show that gains from video captioning can transfer to video QA. An MrDPO-enhanced captioner yields a higher-quality SFT corpus, and models trained on it (video-SALMONN-2+) achieve new open-source SOTA results across multiple video-QA benchmarks.

## 2 RELATED WORK

### 2.1 MULTIMODAL LLMS

Following the adapter-based paradigm that connects multimodal encoders to LLMs, a broad family of models has emerged. For images, LLaVA applies instruction tuning to boost zero-shot performance (Liu et al., 2024b;a; Wei et al., 2022); BLIP-2 links a frozen vision encoder to an LLM via the Q-Former (Li et al., 2023); VILA studies pre-training strategies with strong video-QA transfer (Lin et al., 2023); and InternVL scales visual encoders for stronger image representations (Chen et al., 2023). For silent video, Video-LLaVA aligns image- and video-specific adapters to learn unified representations (Lin et al., 2024); ShareGPT4Video uses GPT-4 to generate dense captions to improve data quality (Chen et al., 2024); and LLaVA-Hound introduces DPO to enhance video understanding (Zhang et al., 2024a). More recent systems expand capability and scale: LLaVA-OneVision strengthens open-source multimodal LLMs across single-/multi-image and silent-video settings with strong image-to-video transfer (Li et al., 2024); LLaVA-Video constructs a high-quality synthetic corpus to push video understanding further (Zhang et al., 2024b); Qwen2-VL and Qwen2.5-VL map variable-resolution images to variable token counts and adopt rotary position embeddings for improved video modelling (Wang et al., 2024a; Bai et al., 2025); NVILA scales spatial/temporal resolution before compression (Liu et al., 2024c); AuroraCap targets video captioning from multiple perspectives and reports favourable results on the proposed VDC benchmark (Chai et al., 2025); and F-16 improves performance via high-frame-rate dense sampling at 16 frame per second (FPS) (Li et al., 2025).

In the realm of audio perception, SALMONN (Tang et al., 2024b) uses a dual-encoder structure and can perform zero-shot audio reasoning tasks. LTU (Gong et al., 2024) and LTU-AS (Gong et al., 2023) trained on a large audio event QA dataset can answer open-ended questions about audio content. Qwen-Audio (Chu et al., 2023) and Qwen2-Audio (Chu et al., 2024) are built on large amounts of audio data to achieve high performance on a wide range of carefully selected audio tasks. Other works (Zheng et al., 2024; Tang et al., 2024c) extend the LLM to perceive spatial audio information obtained from microphone array recordings.

As the visual frame sequence is often paired with audio in real-world video recordings, some studies investigate understanding non-silent video. Vid2Seq (Yang et al., 2023) utilises speech transcriptions to enhance video captioning. video-SALMONN (Sun et al., 2024) uses a multi-resolution causal

Q-Former to understand audio and video simultaneously. video-SALMONN-o1 (Sun et al., 2025) enhances audio-visual reasoning abilities through process DPO. The Google Gemini model achieves video understanding as a native multimodal LLM built upon text, audio, and visual tokens (Team et al., 2024). AVicuna (Tang et al., 2024d) achieves audio-visual temporal understanding by introducing pseudo-untrimmed video data. Video-LLaMA (Zhang et al., 2023) and Video-LLaMA 2 (Cheng et al., 2024) directly concatenate audio and visual tokens for joint audio and video understanding. InternVideo2 (Wang et al., 2024b) aligns video to audio events, speech, and text through cross-modal contrastive learning for joint audio-video understanding. Qwen2.5-Omni (Xu et al., 2025a) not only achieves audio-visual understanding but is also able to generate text and speech in a streaming manner. Qwen3-Omni (Xu et al., 2025b) further enhances omni understanding by supporting longer audio input, enabling multimodal thinking mode, and significantly reducing streaming latency.

## 2.2 RL FOR LLMS

RL with human feedback (RLHF) (Ouyang et al., 2022) is a widely adopted strategy for improving the performance of textual LLMs. Early approaches commonly employed proximal policy optimisation (Schulman et al., 2017) in conjunction with a reward model trained on human preference data. Building on this foundation, DPO (Rafailov et al., 2024) eliminates the need for a separate reward model by leveraging the LLM itself to optimise directly from paired preference data. Iterative RPO (Pang et al., 2024) combines DPO and NLL loss and iteratively conducts optimisation for better reasoning capability. KTO (Ethayarajh et al., 2024) further simplifies the process by removing the requirement for paired preference data altogether. Extending this direction, RL with AI feedback (Lee et al., 2023) adopts a cost-efficient framework by using model-generated feedback in place of human input, significantly reducing human involvement. Most recently, GRPO (Shao et al., 2024) improves efficiency and stability, particularly in tasks such as mathematical reasoning, by comparing relative rewards among candidate responses within a group, thereby removing the need for a separate value network.

## 3 METHODS

### 3.1 MODEL ARCHITECTURE

The overall architecture of our model is illustrated in Fig. 1. The paired sequences of audio and visual frames from each video are fed into the audio and visual encoders separately. Users can provide textual prompts to guide the model in performing specific tasks based on the video content. This structure is implemented by incorporating a separate audio encoder branch to a pre-trained visual LLM, which enables the model to process and understand paired audio-visual sequences without degrading its visual performance.

Specifically, the model accepts a synchronised audio and video stream as its primary input. The video input is first sampled into discrete frames. Each frame is processed independently by a visual encoder to extract salient feature representations. Following this, a visual aligner module maps these high-level visual features into the LLM's input space, transforming them into a sequence of visual tokens. In a parallel process, the accompanying audio waveform is fed into an audio encoder to capture its corresponding features. Similar to the visual branch, an audio aligner then projects these audio features into the same LLM input space, producing a sequence of audio tokens. Capitalising on the inherent temporal alignment of audio and video signals, the audio tokens and visual tokens are arranged in an interleaved manner. In particular, tokens corresponding to the same one-second segment of the input are grouped together, ensuring that the visual and auditory information for any given moment are adjacent in the sequence. The LLM's position embeddings are then applied to this combined sequence, creating the final audio-visual token stream that is fed into the model.

In addition to the audio-visual stream, the model takes a user-provided text prompt as input. This prompt is converted into text tokens using the LLM's text embeddings. These text tokens are then combined with the interleaved audio-visual token sequence, forming the complete input for the LLM. Finally, the text-based backbone LLM is required to generate a text response $\hat{\mathbf{Y}}$ given the user's text prompt $\mathbf{P}$ and the audio-visual token sequence $\mathbf{H}$:

$$\hat{\mathbf{Y}} = \arg\max_{\mathbf{Y}} P(\mathbf{Y}|\mathbf{P}, \mathbf{H}). \tag{1}$$

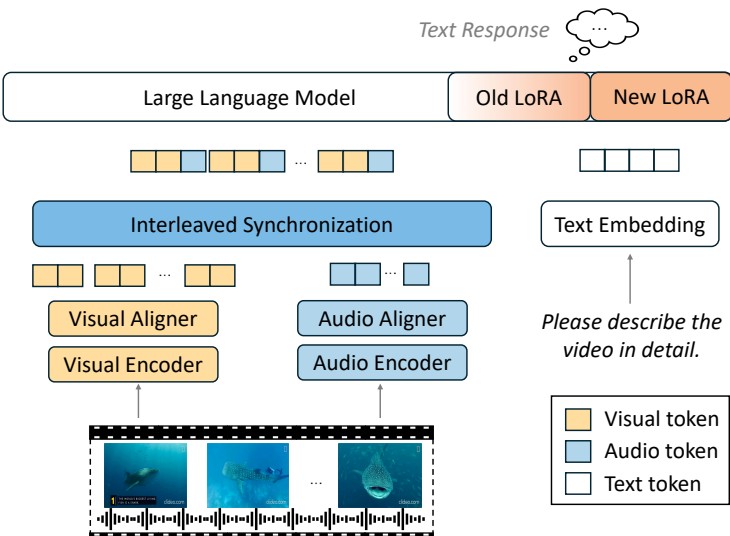

Figure 1: The architecture of video-SALMONN 2. The input video is processed by separate visual and audio branches, which extract visual and audio tokens from the corresponding frame sequences. These tokens are then synchronously interleaved and combined with tokens from the text prompt to form the input to the LLM backbone. In the MrDPO stage, a LoRA proxy is re-initialised, trained, and merged into the LLM backbone at the end of each training round.

## 3.2 MULTI-STAGE TRAINING FOR AUDIO–VISUAL CAPTIONING

To equip the visual LLM with strong video captioning capabilities, we adopt a multi-stage training strategy that enables effective utilisation of audio information for video understanding while preserving the model's visual processing performance. Starting from a well-trained visual LLM, the training pipeline mainly consists of three stages: audio modality alignment, audio-visual SFT, and RL via the proposed MrDPO method. The backbone LLM, the visual encoder, and the audio encoder are kept frozen during training to prevent catastrophic forgetting.

**Audio modality alignment** extends the visual LLM by adding a parallel audio branch, enabling auditory perception abilities. During this stage, only the audio aligner is trained on a large audio dataset, while the rest of the model remains frozen to preserve its original visual understanding performance. Speech recognition and audio captioning are used in the training with the cross-entropy loss function based on the reference speech transcriptions and audio captions.

**Audio-visual SFT** follows audio modality alignment and uses annotated videos to train the model for synchronised, integrated audio–visual understanding over interleaved audio and visual token sequences. We optimise with cross-entropy loss, using video descriptions as ground-truth labels that capture both modalities. To strengthen the backbone LLM's handling of paired sequences, we attach a LoRA adapter and train it jointly (while keeping the encoders frozen). In parallel, the audio aligner is further trained to project audio-encoder outputs into the LLM's input space, ensuring seamless interpretation of audio tokens by the backbone.

**RL based on MrDPO** is applied after audio-visual SFT to address issues such as missing information and hallucinations. Further details are discussed in Section 3.3.

## 3.3 REINFORCEMENT LEARNING WITH MULTI-ROUND DPO

### 3.3.1 A METRIC BASED ON ATOMIC EVENTS

We aim to leverage DPO to enhance the quality of video captions generated by the model. As for very detailed video captions, classical metrics such as BLEU and ROUGE-L are no longer able to measure the quality of the captions. To establish an effective evaluation method for caption completeness and accuracy, we propose using atomic events as an intermediary, enabling artificial intelligence (AI)-driven feedback to automatically assess and refine caption preferences. This approach guides

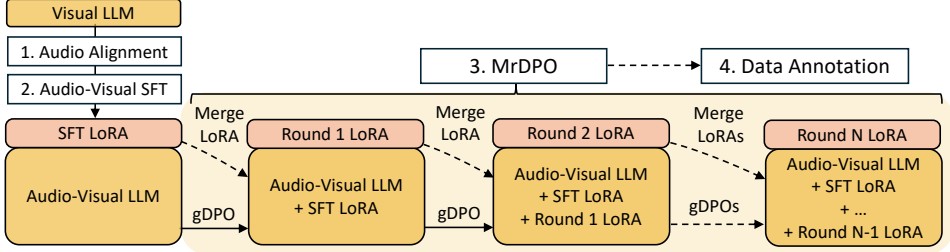

Figure 2: An overview of the training process, including audio modality alignment, audio-visual SFT, and MrDPO. LoRA is introduced during the audio-visual SFT stage. In each round of the MrDPO training, a new LoRA proxy is added while the old LoRA is merged into the LLM, so the model always contains only one activated LoRA. The MrDPO-trained model can be applied to annotate video captions.

the model toward generating more accurate and detailed video captions, ultimately achieving an automated RL training framework based on AI feedback.

Before DPO training, ground-truth video captions are first decomposed into atomic events using a powerful text LLM, such as GPT-4o, to provide references for evaluating the captions generated by the model. Next, we apply the nucleus sampling method (Holtzman et al., 2020) to generate caption pairs from the model's output distribution for the input video.

The atomic events from the ground-truth video captions are used to identify the preferred sample within each pair. Specifically, we input a video caption and the corresponding list of ground-truth atomic events into a powerful text LLM like GPT-3.5. The LLM is prompted to identify which atomic events from the ground-truth list are missing in the caption, which events from the ground-truth list are described incorrectly, and which events described in the caption are absent from the ground-truth list. The missing ones are referred to as **"missing events"**, and the latter two types of events are referred to as **"hallucination events"**, respectively. The missing and hallucination rates for each caption are calculated by dividing the number of each type of event by the total number of atomic events in the ground-truth caption. The total error rate of a caption is then obtained by summing its missing and hallucination rates.

In each caption pair, the one with a lower total error rate is regarded as the preferred sample in DPO. To improve efficiency and mitigate the impact of LLM evaluation noise, caption pairs with small differences in these metrics are excluded from the DPO training set.

### 3.3.2 MULTI-ROUND DPO WITH LORA PROXY

Unlike previous approaches that applied only single-round DPO to multimodal LLMs, we introduce a multi-round strategy, as prolonged offline training with a single round fails to optimise the model effectively due to the reference model being biased against the most recent model update in the DPO algorithm. In the multi-round framework, at each $t$th round, the following steps are taken to perform DPO training for the current round.

1. First, pre-trained LoRA module $\Delta_{t-1}$ is merged into the LLM backbone $\Lambda_{t-1}$ to derive a new LLM backbone $\Lambda_t$ that is equivalent to $\Lambda_{t-1}$ with $\Delta_{t-1}$, based on Eqn. (2):

$$\mathbf{W}_t = \mathbf{W}_{t-1} + \alpha \, \mathbf{A}_{t-1} \mathbf{B}_{t-1}, \tag{2}$$

where $\mathbf{W}_t$ and $\mathbf{W}_{t-1}$ are the weight parameters to adapt in $\Lambda_t$ and $\Lambda_{t-1}$, $\alpha$ is the scaling factor of LoRA, $r$ is the rank of LoRA, $d$ is the dimension of $\mathbf{W}_{t-1}$. $\mathbf{A}_{t-1} \in \mathcal{R}^{d \times r}$ and $\mathbf{B}_{t-1} \in \mathcal{R}^{r \times d}$ are the low-rank matrix parameters of LoRA in the previous round $t-1$, and $\mathbf{W} \in \mathcal{R}^{d \times d}$ is the parameter of LLM backbone.

2. Next, a newly initialised LoRA module, $\tilde{\Delta}_t$, is added to the LLM backbone, forming the new policy model $\Lambda_t$ for round $t$. During this round, only the LoRA parameters $\tilde{\Delta}_t$, referred to as the LoRA proxy, are updated, while all other LoRA parameters remain fixed. To mitigate the growing discrepancy between the reference and policy models caused by freezing the reference model in standard DPO, $\Lambda_t$ is adopted as the updated reference model for round $t$.

3. At last, $\tilde{\Delta}_t$ is trained to obtain $\Delta_t$, which can be achieved using the standard DPO loss. However, after multiple training rounds, the model is prone to getting stuck in local optima, leading to

stagnant performance. To alleviate this issue by stabilising the training, a guided DPO (gDPO) loss is proposed as

$$\mathcal{L}_{\text{gDPO}}(\pi_\theta; \pi_{\text{ref}}) = - \mathbb{E}_{(\mathbf{x}, \mathbf{y}_{\text{win}}, \mathbf{y}_{\text{lose}}) \sim \mathcal{D}} \left[ \log \sigma \left( \beta \log \frac{\pi_\theta(\mathbf{y}_{\text{win}} \mid \mathbf{x})}{\pi_{\text{ref}}(\mathbf{y}_{\text{win}} \mid \mathbf{x})} - \beta \log \frac{\pi_\theta(\mathbf{y}_{\text{lose}} \mid \mathbf{x})}{\pi_{\text{ref}}(\mathbf{y}_{\text{lose}} \mid \mathbf{x})} \right) \right]$$
$$+ \lambda \, \mathbb{E}_{(\mathbf{x}, \mathbf{y}_{\text{gt}}) \sim \mathcal{D}_{\text{gt}}} \log \pi_\theta(\mathbf{y}_{\text{gt}} | \mathbf{x}), \tag{3}$$

where $\pi_\theta = \{\Lambda_t, \tilde{\Delta}_t\}$ and $\pi_{\text{ref}} = \Lambda_t$ represent the policy and reference models for round $t$ respectively, $\sigma$ is the sigmoid function, and $\beta$ is a hyper-parameter controlling the deviation from $\pi_{\text{ref}}$. Variables in the first term follow the definitions in the standard DPO loss (Rafailov et al., 2024). In the second term corresponding to cross-entropy learning towards the ground-truth video captions, $\lambda$ is the weight of the second term in the overall loss, $\mathcal{D}_{\text{gt}}$ denotes the SFT training dataset, and $(\mathbf{x}, \mathbf{y}_{\text{gt}})$ corresponds to a video and its paired ground-truth text description. $\mathbf{y}_{\text{win}}$ and $\mathbf{y}_{\text{lose}}$ are preferred and dispreferred video captions generated by $\Lambda_t$ for $\mathbf{x}$, which are judged using a text LLM based on the atomic-event-based metric proposed in Section 3.3.1. Each mini-batch of training samples is randomly selected from $\mathcal{D}_{\text{gt}}$. It is worth noting that this gDPO loss is different from Iterative RPO (Pang et al., 2024), we use the guidance of ground truth instead of the chosen sample, which aims to stabilize training instead of avoiding probability decrease.

## 3.4 Transferring Caption Optimisation Gains to Video QA

Following the application of MrDPO, a significant enhancement in the model's video captioning performance can be achieved. However, we found that the model's general video understanding capabilities are almost entirely determined by audio-visual SFT. The subsequent RL training, which focuses exclusively on captioning, did not yield further improvements in the model's general video understanding capabilities.

In the audio-visual SFT stage, the quality and nature of the training data play a fundamental role. Typically, this data comprises two main categories: video caption data and video QA data. We posit that video caption data is more essential of the two, especially for larger and stronger models. High-quality video captions enable the model to develop a holistic understanding of a video, encompassing its content, intricate details, underlying themes, and emotional tone. In contrast, video QA data serves to refine the model's ability to accurately apply the knowledge it has acquired from the caption data to specific tasks.

A significant limitation of existing video caption data is that they are largely generated by existing models. While these captions can provide a general description of the video's content, they often lack the requisite detail, which can result in models trained on this data failing to accurately perceive subtle but important aspects of the video. Consequently, we utilise the MrDPO-trained model to re-annotate existing video data, thereby creating a new, higher-quality set of video captions. We then apply this new caption data and the original QA data to directly perform audio-visual SFT after audio alignment. Through this strategy, we aim to transfer the model's powerful video captioning performance to a more generalised and robust video understanding capability.

## 4 Experimental Setup

### 4.1 Model Specifications

video-SALMONN 2 series consists of video-SALMONN 2 (7B), video-SALMONN $2_{\text{F-16}}$ (7B), and video-SALMONN 2+. video-SALMONN 2 (7B) is built on an internally trained high-performance visual LLM, which is further fine-tuned on LLaVA-OneVision-7B (Li et al., 2024). The model processes video frames at a per-second frame rate of 1, and can handle up to 110 frames with $384 \times 384$ pixels per frame. video-SALMONN $2_{\text{F-16}}$ (7B) is trained based on F-16 (Li et al., 2025) and processes 16 FPS video up to 1760 frames with $384 \times 384$ pixels. video-SALMONN 2+ series contains 3B, 7B, and 72B versions, all derived from Qwen 2.5-VL series. The model processes video frames at 10 FPS, and can handle up to 768 frames with 61250 pixels per frame.

For the audio branch, we use the Whisper-Large-v3 encoder (Radford et al., 2023) as the audio encoder, and a window-level Q-Former (Tang et al., 2024a) with a window length of 0.5 seconds as the audio aligner, which produces 60 audio tokens for a 30-second input. The rank $r$ and scaling factor $\alpha$ of LoRA are set to 128 and 2.0, respectively. During training, the visual encoder, the audio encoder, and the LLM remain frozen.

## 4.2 DATA AND TRAINING SPECIFICATIONS

All models for the video-SALMONN 2 series are trained with modality alignment. video-SALMONN 2 7B is trained with audio-visual SFT and MrDPO. video-SALMONN 2 (F-16) 7B is trained with data annotated with video-SALMONN 2 7B. video-SALMONN 2+ series is trained with data annotated with a Qwen2.5-VL 7B-based MrDPO model.

In the audio modality alignment stage, LibriSpeech-960h (Panayotov et al., 2015) and AudioCaps (Kim et al., 2019) are used to train the audio aligner. LibriSpeech-960h is utilized for speech recognition training, while AudioCaps is employed for audio captioning training. In the audio-visual SFT stage, experiments are performed using FineVideo (Farré et al., 2024), CinePile (Rawal et al., 2024), and about 13k videos with rich audio information from LLaVA-Video-178k (Zhang et al., 2024b). Both video captioning and video QA are trained during audio-visual SFT.

In the MrDPO stage, before each training round, the model generates a pair of captions for each video in the SFT dataset. To determine whether a caption pair is suitable for DPO, we evaluate the missing information rate and hallucination rate using GPT-3.5. It is also feasible to use a smaller model for evaluation, as shown in Appendix C. If deemed suitable, one caption is selected as the preferred sample, while the other is rejected. The selection criteria for each round are detailed in Appendix D. We explore the values of $\lambda$ in Eqn. (3) in Appendix E and finally set it to 0.1 throughout the entire MrDPO stage. Since only video captioning is trained during MrDPO, the video QA performance of the model will slightly decline during DPO rounds, and QA data is added in the final round. Detailed training cost can be found in Appendix B.

For further data annotation, 100k videos are randomly selected from the training datasets and re-annotated.

For captioning evaluation, we curated an open-sourced video captioning benchmark[1] to evaluate the event missing rate (Miss) and hallucination rate (Hall). Details of the test data can be found in Appendix F. The benchmark consists of 483 carefully selected videos, each labelled with complete audio-visual captions by human annotators. Atomic events for the test dataset were initially obtained using GPT-4o and then manually refined. GPT-3.5 is used to evaluate the event missing and hallucination rates of the generated captions. Details are shown in Appendix G.

## 5 EXPERIMENTAL RESULTS

### 5.1 CAPTIONING EVALUATION RESULTS AND ANALYSIS

Table 1: Captioning evaluation and ablation on video-SALMONN 2 (7B).

(a) The comparison of captioning performance of the models. VMME denotes QA results of GPT-4o on Video-MME based on video descriptions generated by the corresponding models.

| Model | Our Caption Benchmark | | | VDC$_{Detailed}$ | VMME |
|---|---|---|---|---|---|
| | %Miss↓ | %Hall↓ | %Total↓ | %Acc↑\|Score↑ | %Acc↑ |
| GPT-4o | 17.0 | 14.2 | 31.2 | 46.3\|2.5 | 64.3 |
| Qwen2.5-VL (7B) | 21.9 | 17.4 | 39.2 | 44.5\|2.4 | 55.0 |
| InternVideo 2.5 (7B) | 30.8 | 15.0 | 45.8 | 39.6\|2.2 | 51.8 |
| VideoLLaMA 3 (7B) | 44.9 | **11.6** | 56.5 | 33.4\|1.9 | 46.3 |
| Qwen2.5-Omni (7B) | 26.7 | 21.7 | 48.1 | 39.7\|2.2 | 52.7 |
| video-SALMONN 2 (7B) | **10.0** | 12.9 | **22.9** | **46.1\|2.5** | **65.9** |

(b) The component-level ablation study conducted with video-SALMONN 2 (7B) for MrDPO.

| Method | Total%↓ | %Improve |
|---|---|---|
| Visual | 50.7 | - |
| + SFT | 41.8 | +17.6 |
| + DPO | 37.8 | +9.6 |
| + gDPO | 39.7 | -5.0 |
| + LoRA Proxy | 33.7 | +15.1 |
| + MrDPO | 22.9 | +32.0 |

To demonstrate the powerful captioning capability of video-SALMONN 2 and the effectiveness of MrDPO, we evaluate video-SALMONN 2 (7B) on our caption benchmark. The performance on the "detailed" subset of the VDC caption benchmark (Chai et al., 2025) is also reported for examining visual captioning capability. Another way to evaluate caption completeness is by using the generated video captions rather than the raw video frames as input to a powerful text-based LLM for the video QA task. We used the Video-MME benchmark (Fu et al., 2025a) for testing. Specifically, we first use captioning models to generate descriptions for each video in Video-MME, and then prompt GPT-4o to answer the relevant questions based solely on these captions. In this setting, higher QA accuracy suggests that the generated captions are more complete and informative. All results are reported in Table 1a. video-SALMONN 2 (7B) outperforms other models in both information missing and hallucination rates on our caption benchmark, as well as VDC detailed and Video-MME captioning. The high correlation for all 3 metrics shows the effectiveness of our caption benchmark.

---

[1]https://huggingface.co/datasets/videoSALMONN2/video-SALMONN_2_testset

Among existing open-source multimodal LLMs, few can provide detailed and accurate video descriptions. The caption quality of Qwen2.5-VL is the best among existing open-source models. Notably, some open-source models, such as VideoLLaMA 3, tend to generate shorter captions, leading to relatively high information missing rates but low hallucination rates. Qwen2.5-Omni can perceive both audio and visual information simultaneously, but still mainly focuses on visual information, often omitting human speech in audio and other sound events in captions. The visual-only version of GPT-4o lacks audio comprehension, leading to a higher rate of missed events. We also perform Elo ranking with human evaluators inspired by Chai et al. (2025), which validates the alignment of our proposed metrics with human perception and highlights the strong video captioning capabilities of video-SALMONN 2. Details are provided in Appendix H.

## 5.2 ABLATION ON MRDPO

To address the inherent instability of multi-round DPO, we employ two key strategies: using the gDPO loss and training with LoRA proxies. A complement component-level ablation study is shown in Table 1b.

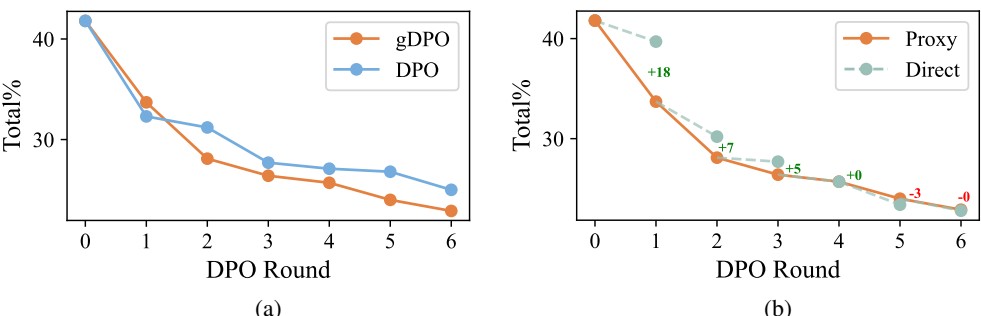

(a)                                                             (b)

Figure 3: SubFig. (a) compares the performances of training with the proposed gDPO loss and the classical DPO loss for six rounds, denoted as "gDPO" and "DPO". SubFig. (b) compares the performance between training a new LoRA proxy in each round and directly fine-tuning the existing LoRA based on the model of the last MrDPO round, denoted as "Proxy" and "Direct". Total error rates on our caption benchmark are evaluated here.

Specifically, Fig. 3 further demonstrates the details on two key strategies. As illustrated in Fig. 3a, while classical DPO shows a slight advantage in the first round, the gDPO loss, which incorporates an additional cross-entropy term conditioned on ground-truth captions, consistently outperforms it from the second round onward, confirming its effectiveness in stabilizing optimization. Moreover, Fig. 3b shows this "Proxy" approach yields better performance than directly fine-tuning the existing LoRA ("Direct"), especially in the early stages. This improvement likely stems from the fact that merging LoRA progressively strengthens the model's foundation by integrating prior adaptations, while re-initializing a new LoRA allows the model to flexibly explore fresh low-rank subspaces aligned with current preference signals, thus avoiding stagnation in outdated parameter spaces.

## 5.3 ABLATION ON SFT DATA

We perform audio-visual SFT on the audio-aligned model using both the original SFT data and the SFT data generated by the model after MrDPO. Specifically, we experimented with training video-SALMONN $2_{\text{F-16}}$ (7B) (Li et al., 2025) using data generated by video-SALMONN 2 (7B); and training video-SALMONN 2+ (72B) using data generated by Qwen 2.5-VL 7B MrDPO model. We then compared these models with those trained using the corresponding original SFT data. The evalua-

Table 2: Results of models using different SFT data on Video-MME. S, M, and L denote short, medium, and long.

| Model | Video-MME | | | |
|---|---|---|---|---|
| | Avg.↑ | S↑ | M↑ | L↑ |
| F-16 (7B) | 65.0 | 78.9 | 63.2 | 52.8 |
| +SFT data | 69.2 | 79.8 | 67.1 | 60.7 |
| +generated data | **70.2** | **80.1** | **68.8** | **61.7** |
| Qwen2.5-VL (72B) | 73.3 | 80.8 | 73.5 | 65.4 |
| +SFT data | 78.8 | 82.3 | 81.2 | **72.8** |
| +generated data | **79.7** | **85.0** | **81.7** | 72.3 |

tion results on Video-MME are shown in Table 2. Models trained with generated data consistently

outperform models trained with original SFT data, which reveals that models enhanced on captioning by MrDPO can be used to generate new data and fine-tune stronger models.

## 5.4 OVERALL RESULTS ON VIDEO QA BENCHMARKS

Table 3: Evaluation results of video-SALMONN 2 series models on video understanding benchmarks. Models denoted with $*$ can perceive audio-visual information, while other models can only understand visual-only frames. The best results for each scale are **in bold**.

| Model | Audio-visual QA | | | | | Visual-only QA | |
|---|---|---|---|---|---|---|---|
| | Video-MME | WorldSense | AVUT | Video-Holmes | DailyOmni | MLVU$_{(Dev)}$ | LVBench |
| VideoLLaMA3 (2B) | 59.6 | - | - | - | - | 65.4 | 41.6 |
| Qwen2.5-Omni (3B)$^*$ | 62.0 | - | - | - | 40.5 | - | - |
| Qwen2.5-VL (3B) | 61.5 | - | - | - | 37.4 | 68.2 | 43.3 |
| video-SALMONN 2+ (3B)$^*$ | **68.3** | **48.3** | **66.2** | **42.2** | **67.7** | **70.5** | **48.6** |
| video-SALMONN (13B)$^*$ | 43.3 | - | 38.3 | - | - | - | - |
| LLaVA-Video (7B) | 63.3 | 40.2 | 56.5 | - | - | 70.8 | - |
| VideoLLaMA2 (7B)$^*$ | 54.9 | 25.4 | 44.9 | - | 35.2 | 32.7 | - |
| VideoLLaMA3 (7B) | 66.2 | - | - | - | - | 73.0 | 45.3 |
| InternVideo2.5 (8B) | 65.1 | - | - | - | - | 72.8 | 46.4 |
| Qwen2.5-Omni (7B)$^*$ | 64.3 | 45.4 | - | 16.4 | 47.5 | - | - |
| Qwen2.5-VL (7B) | 65.1 | - | - | 27.8 | 40.7 | 70.2 | 45.3 |
| video-SALMONN 2 (7B)$^*$ | 67.4 | 48.6 | 65.6 | 40.7 | 66.3 | 68.3 | - |
| video-SALMONN 2$_{F-16}$ (7B)$^*$ | 70.2 | 49.6 | 66.3 | 41.6 | 66.8 | 69.6 | - |
| video-SALMONN 2+ (7B)$^*$ | **73.4** | **50.9** | **69.5** | **46.9** | **71.8** | **73.6** | **49.7** |
| GPT-4o | 71.9 | 42.6 | 56.6 | 42.0 | 56.5 | 64.6 | 30.8 |
| Gemini-1.5 Pro$^*$ | 75.0 | 48.0 | **78.3** | 41.2 | - | - | 33.1 |
| Qwen3-Omni-Flash$^*$ | 71.4 | 54.1 | - | 57.3 | 76.2 | 75.5 | 51.1 |
| LLaVA-Video (72B) | 70.5 | - | - | - | - | 74.4 | - |
| VideoLLaMA2 (72B)$^*$ | 61.4 | - | - | - | - | 61.2 | - |
| Qwen2.5-VL (72B) | 73.3 | - | - | 50.2 | 61.8 | 74.6 | 47.3 |
| video-SALMONN 2+ (72B)$^*$ | **79.7** | **56.5** | 72.2 | **57.8** | **79.4** | **80.4** | **55.5** |

The overall evaluation results of video-SALMONN 2 series on general video understanding benchmarks are presented in Table 3. To comprehensively evaluate the performance of video-SALMONN 2 series, we select leading models among ∼3B, ∼7B, and larger scales as our baselines. Both open-source video models, including VideoLLaMA3 (Zhang et al., 2025), LLaVA-Video (Zhang et al., 2024b), InternVideo 2.5 (Wang et al., 2025b), and Qwen2.5-VL (Bai et al., 2025), and the leading closed-source model GPT-4o are evaluated. To highlight the advantages of our model among audio-visual models, video-SALMONN (Sun et al., 2024), VideoLLaMA 2 (Cheng et al., 2024), Qwen 2.5-Omni (Xu et al., 2025a), Qwen3-Omni-Flash (Xu et al., 2025b), and Gemini-1.5 Pro (Team et al., 2024) are also compared. Audio-visual QA benchmarks include Video-MME (Fu et al., 2025a), WorldSense (Hong et al., 2025), AVUT (Yang et al., 2025b), Video-Holmes (Cheng et al., 2025), and DailyOmni (Zhou et al., 2025b). Visual-only benchmarks include MLVU (Zhou et al., 2025a) and LVBench (Wang et al., 2025a).

For audio-visual benchmarks, video-SALMONN 2+ shows dominating performance and outperforms all models with similar scales for both visual-only models and audio-visual models, even competitive to larger models. video-SALMONN 2+ (3B) outperforms all existing 7B models, and video-SALMONN 2+ (7B) is competitive to existing 72B models. The largest video-SALMONN 2+ (72B) even consistently performs better than GPT-4o and Qwen3-Omni-Flash, while surpassing Gemini-1.5 Pro on most of the audio-visual QA benchmarks. For visual-only video understanding, video-SALMONN 2+ still retains strong capabilities, achieving slight improvements compared to the base model Qwen2.5-VL, and remains highly competitive against same-scale visual-only state-of-the-art models.

## 6 CONCLUSIONS

We introduced video-SALMONN-2, a family of high-performance audio–visual LLMs for video captioning and video QA. Our core method, MrDPO, addresses DPO reference staleness to enable continual self-improvement, producing captions that are richer and more faithful than GPT-4o and Gemini-1.5 Pro. Using this strengthened captioner to re-annotate video data yields higher-quality supervision for SFT, and we show that gains from captioning transfer to general video understanding. Across standard video-QA benchmarks, our 3B/7B models lead their size class, and the 72B model

surpasses all open-source competitors. Code, models, and data will be released to support further research.

## 7 REPRODUCIBILITY STATEMENT

We provide detailed descriptions of the model architecture, training steps, training settings, and datasets in Section 3 and Section 4. All non-public datasets, code, and final models will be released upon acceptance, which provides enough reproducibility for the work.

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

## A  THE USE OF LARGE LANGUAGE MODELS

We utilized Gemini-2.5-Pro and GPT-5 to assist us in checking for grammar errors and polishing the fluency of our sentences.

## B  TRAINING COST FOR VIDEO-SALMONN 2 (7B)

Regarding training settings, the resource consumption of video-SALMONN 2 (7B) for each training step is shown in Table 4. video-SALMONN 2+ (7B) is trained with similar settings. In our experiments, we run a total of six gDPO rounds in MrDPO.

Table 4: Resource consumption in each training stage.

| Training Stage | Used GPUs | Batch Size / GPU | Updates | Training Time |
|---|---|---|---|---|
| Audio Modality Alignment | $32 \times$ H800s | 8 | 30,000 | 3 hours |
| Audio-Visual SFT | $32 \times$ H800s | 1 | 15,475 | 14 hours |
| MrDPO Round 1-5 | $8 \times$ H800s | 1 | 1,000 | 2 hours |
| MrDPO Round 6 | $32 \times$ H800s | 1 | 841 | 1 hours |

## C  TRAINING WITH MODELS WITH SMALLER SCALE AS EVALUATOR

We also try using Qwen3-4B Yang et al. (2025a) as the LLM for evaluating the quality of video captions during training, and still use GPT-3.5 for testing. All prompts, generation settings, and training settings remain the same with the GPT-3.5 version. The results of using these two models as evaluators are similar, as shown in Table 5.

Table 5: The captioning results of using Qwen3-4B and GPT-3.5 as evaluators in the first gDPO round.

| Evaluation LLM during training | Miss%↓ | Hall%↓ | Total%↓ |
|---|---|---|---|
| GPT-3.5 | 13.8 | 19.9 | 33.7 |
| Qwen3-4B | 14.3 | 19.4 | 33.7 |

## D  SAMPLES SELECTING CRITERIA IN MRDPO

To achieve better performance and training efficiency, we take a specially designed strategy to select proper preference pairs. A sample pair is selected if one sample is significantly better than the other. We consider the total error rate $\Delta e$ as the main metric. Besides, to avoid repeatedly decoding, we also consider the repetition rate $\Delta r$ of each sample. Table 6 shows the threshold used in each round of MrDPO.

Table 6: The data selecting threshold used in each gDPO round. A negative number means that the chosen sample can be worse than the rejected sample in this metric to some degree.

| gDPO Round | Threshold Used | |
|---|---|---|
| | $\Delta e$ | $\Delta r$ |
| 1 | $\geq 5\%$ | $\geq 1\%$ |
| 2 | $\geq 20\%$ | $\geq -1\%$ |
| 3 | $\geq 23\%$ | $\geq -1\%$ |
| 4 | $\geq 23\%$ | $\geq -1\%$ |
| 5 | $\geq 23\%$ | $\geq -1\%$ |
| 6 | $\geq 23\%$ | $\geq -1\%$ |

# E    HYPERPARAMETER SEARCH IN gDPO

In gDPO, the weight $\lambda$ of the cross-entropy regularization term is an important hyperparameter. We simply performed a search over its values in the first round of MrDPO. Results are shown in Table 7.

Table 7: Results of the first round of MrDPO training with different $\lambda$.

| $\lambda$ | Our Caption Benchmark | | |
|---|---|---|---|
| | %Miss↓ | %Hall↓ | %Total↓ |
| 0 | 13.7 | 18.6 | 32.3 |
| 0.01 | 13.7 | 19.0 | 32.8 |
| 0.1 | 13.8 | 19.9 | 33.7 |
| 1 | 15.4 | 21.9 | 36.2 |
| 10 | 16.8 | 22.8 | 39.7 |

As the results show, when $\lambda$ is greater than 1, the weight of the cross entropy loss becomes too large and hinders the optimization of the DPO loss. On the other hand, although the model trained with small $\lambda$ (less than 0.01) seems better in the first gDPO round, the cross-entropy term is too small to stabilise MrDPO training, as Figure 3a shows. Therefore, we finally choose 0.1 as a suitable value of $\lambda$.

# F    ABOUT THE TEST DATASET

Figure 4 shows the video type distribution and the video length distribution of our caption benchmark. The benchmark covers 14 different fields. All the videos are between 30s to 60s, with an average duration of 51s. Table 8 shows more statistics about the labelled captions and atomic events of the benchmark. Specifically, there are 6.1 audio-related atomic events per video, including 4.6 speech-related events and 1.5 non-speech events.

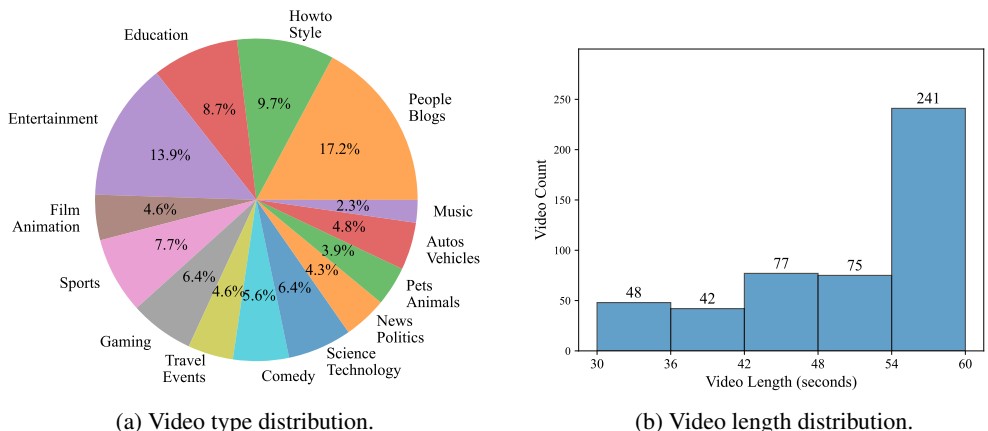

(a) Video type distribution.          (b) Video length distribution.

Figure 4: The basic information of our benchmark data.

Table 8: Statistics about the labelled captions and atomic events of the benchmark.

| #Sample | #Vocabulary | #Word | Average Statistics | |
|---|---|---|---|---|
| | | | Caption Length | #Atomic Event |
| 483 | 17137 | 296,938 | 615 words | 34.2 |

We also evaluated the effect of input modalities for video-SALMONN 2 (7B) on our benchmark. We have evaluated the base visual model, visual-only input, audio-only input, and audio-visual input.

The results are shown in Table 9. After rounds of DPO, video-SALMONN 2 (7B)$_V$ significantly reduces hallucination rate, although audio is not given. The missing rate is still a bit high due to the lack of audio. For the audio-only version, since most of the atomic events are related to visual information, the missing rate is significantly higher than others. The audio-visual model gets the best result through comprehensive information and complete audio-visual understanding.

Table 9: Results of different input modalities for video-SALMONN 2 (7B).

| Model | Our Caption Benchmark | | |
|---|---|---|---|
| | %Miss↓ | %Hall↓ | %Total↓ |
| Visual Base Model | 23.3 | 27.4 | 50.7 |
| video-SALMONN 2 (7B)$_V$ | 21.4 | 14.1 | 35.4 |
| video-SALMONN 2 (7B)$_A$ | 62.6 | 22.0 | 84.0 |
| video-SALMONN 2 (7B)$_{AV}$ | **10.0** | **12.9** | **22.9** |

## G  EVALUATION PROCESS ON OUR CAPTION BENCHMARK

During testing, GPT-3.5 is used as LLM for evaluating the quality of video captions. We input the captions generated by the model and the list of groundtruth atomic events into GPT-3.5, and ask GPT-3.5 to list the following three types of events and provide the quantity of each:

1. **Missing Events**: Atomic events from the ground-truth list that are missing in the caption.
2. **Incorrect Events**: Atomic events from the ground-truth list that are described incorrectly in the caption.
3. **Hallucination Events**: Events described in the caption that are absent from the ground-truth atomic event list.

Note that incorrect events and hallucination events both belong to the model's "hallucination" and will be used to calculate the hallucination rate.

The quotient between the number of missing events and the number of all atomic events is the missing rate, and the quotient between the number of incorrect and hallucination events and the number of all atomic events is the hallucination rate. The total error rate is the sum of the missing and hallucination rates.

The prompts for testing can be referenced from the scripts provided in the open-source test data.

## H  ELO RANKING RESULTS

We perform the Elo ranking by human to rate the video captioning capability of Gemini-1.5 Pro, GPT-4o, our visual base model, and the final video-SALMONN 2. Parameters of the Elo ranking system are provided in Table 10.

Table 10: Parameters of the Elo ranking system.

| Parameter | Value |
|---|---|
| Initial ELO mean | 1000 |
| Base of logarithm | 10 |
| Scaling factor | 400 |
| K-factor | 8 |

We collected 180 pairs of captions generated by different models as simulated matches. Annotators are provided with the video and two corresponding video captions generated by two different models, respectively, and they are expected to select the better caption according to the completeness and correctness of the caption. The final Elo rating results and the total error rates on our benchmark of each model are provided in Table 11.

Table 11: Elo rating results compared with the total error rates of different models. "Total" represents the total error rate of video captioning on our caption benchmark.

| Model | Total%↓ | Elo Rating↑ |
|---|---|---|
| GPT-4o Visual | 31.2 | 976 |
| Gemini-1.5 Pro | 38.3 | 964 |
| Ours-Visual Base | 50.7 | 945 |
| video-SALMONN 2 | **22.9** | **1115** |

## I AN EXAMPLE OF ATOMIC EVENTS

We show an example of atomic events of a randomly selected video in Fig. 5.

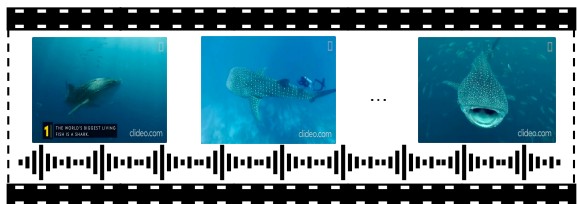

**Atomic Event List**

1. The video begins with a whale shark swimming in the ocean.
2. The whale shark moves slowly through the water.
3. The number '1' in a yellow box apeears in the video.
4. The text 'THE WORLD'S BIGGEST LIVING FISH IS A SHARK.' appears in the video.
5. A vibrant coral reef teeming with small fish appears.
6. A woman says, "The world's biggest living fish is a shark. Of the estimated 34,000 species of fish, the largest are whale sharks."
7. Background music can be heard.
8. A diver swims near the whale shark.
9. The scene changes again, showing a diver swimming near the whale shark.
10. A woman's voice says, "These gentle giants usually grow to about 40 feet long and weigh an estimated 15 tons. Their mouths alone can span 4 feet wide."
11. The background music continues.
12. Text 'clideo.com' can be seen in the bottom right corner.
13. The scene changes, showing a close-up view of the whale shark's mouth.

Figure 5: An example of the atomic event list of a video.

