# OpenReview forum: "video-SALMONN 2: Caption-Enhanced Audio-Visual Large Language Models"
_ICLR.cc/2026/Conference — Submitted to ICLR 2026_

### Official Review · Reviewer_S8iw · 2025-10-29

**Soundness:** 3
**Presentation:** 2
**Contribution:** 3
**Rating:** 6
**Confidence:** 3

**Summary:**

The paper presents Video-SALMONN 2, a family of audio-visual large language models that achieve state-of-the-art results in video captioning and question answering through multi-round direct preference optimization (MrDPO) with a caption-quality objective.
MrDPO continually refreshes training references via lightweight adapters, producing more detailed and accurate captions that transfer effectively to video-QA tasks, surpassing proprietary systems like GPT-4o and Gemini-1.5 Pro.

**Strengths:**

•	The paper introduces an original multi-round DPO (MrDPO) framework that continuously updates reference models to prevent staleness, leading to more accurate and faithful video captioning.
	•	It demonstrates strong methodological quality and clarity, effectively combining reinforcement-based preference optimization with lightweight adapters for stable and scalable training.
	•	Extensive experiments across diverse benchmarks show consistent SOTA performance in both captioning and video-QA, highlighting the method’s strong generalization and practical impact.

**Weaknesses:**

See questions.

**Questions:**

To ensure fair evaluation, the paper should also report results on long-video benchmarks without audio input to distinguish improvements from the proposed training strategy versus those merely due to the additional audio modality.

---

> ### Author Response · Authors · 2025-11-21
> **Response to Reviewer S8iw**
>
> Both MLVU and LVBench are evaluated without audio input, yet the video-SALMONN 2 series continues to achieve strong performance relative to models of similar size.
>
> We sincerely thank Reviewer S8iw for the thoughtful comments, constructive suggestions, and positive assessment. We have addressed this point in detail in our rebuttal, including additional analysis clarifying the effect of audio versus training strategy. Please let us know if you have any further concerns, we would be more than happy to clarify them. If all your questions have been resolved, we would be grateful if you could kindly reconsider our work in your final evaluation. Thank you again for your time and valuable feedback.

---

> > ### Comment · Reviewer_S8iw · 2025-11-22
> >
> > Thanks for the clarification! I have just one small question: could you provide the specific results or analysis that support the statement that “the Video-SALMONN 2 series still outperforms models of similar size even without audio input”?
> > It would be helpful to see a table or comparison experiment on the same benchmark, both with and without audio input.

---

> > > ### Author Response · Authors · 2025-11-23
> > >
> > > Thank you for your quick reply! In the table below, we provide the evaluation results of the video-SALMONN 2+ models under both visual-only and audio-visual input settings. For MLVU and LVBench, the results are identical since these benchmarks do not include audio in the provided video. As shown, under pure visual input, video-SALMONN 2+ already outperforms other visual-only models of similar size to a certain extent; furthermore, when audio is incorporated, the model achieves additional performance gains on audio-visual benchmarks.
> > >
> > > | Model                               | VideoMME | WorldSense | AVUT | VideoHolmes | DailyOmni | MLVU | LVBench |
> > > | ----------------------------------- | -------- | ---------- | ---- | ----------- | --------- | ---- | ------- |
> > > | VideoLLaMA 2 7B                     | 54.9     | 25.4       | 44.9 | -           | 35.2      | 32.7 | -       |
> > > | LLaVA-Video 7B                      | 63.3     | 40.2       | 56.5 | -           | -         | 70.8 | -       |
> > > | Qwen 2.5-VL 7B                      | 65.1     | -          | -    | 27.8        | 40.7      | 70.2 | -       |
> > > | video-SALMONN 2+ 7B (visual-only)   | 67.9     | 42.6       | 57.9 | 43.8        | 57.6      | 73.6 | 49.7    |
> > > | video-SALMONN 2+ 7B (audio-visual)  | 73.4     | 50.9       | 69.5 | 46.9        | 71.8      | 73.6 | 49.7    |
> > > |                                     |          |            |      |             |           |      |         |
> > > | GPT-4o                              | 71.9     | 42.6       | 56.6 | 42.0        | 56.5      | 64.6 | 30.8    |
> > > | VideoLLaMA 2 72B                    | 61.4     | -          | -    | -           | -         | 61.2 | -       |
> > > | LLaVA-Video 72B                     | 70.5     | -          | -    | -           | -         | 74.4 | -       |
> > > | Qwen 2.5-VL 72B                     | 73.3     | -          | -    | 50.2        | 61.8      | 74.6 | 47.3    |
> > > | video-SALMONN 2+ 72B (visual-only)  | 75.0     | 47.4       | 60.2 | 53.0        | 63.1      | 80.4 | 55.5    |
> > > | video-SALMONN 2+ 72B (audio-visual) | 79.7     | 56.5       | 72.2 | 57.8        | 79.4      | 80.4 | 55.5    |

---

### Official Review · Reviewer_8DaU · 2025-10-30

**Soundness:** 2
**Presentation:** 3
**Contribution:** 2
**Rating:** 4
**Confidence:** 3

**Summary:**

The paper presents video-SALMONN 2, a family of audio-visual LLMs that attain state-of-the-art captioning and strong video-QA by (i) introducing Multi-round Direct Preference Optimization (MrDPO)—a preference-learning scheme that repeatedly refreshes the DPO reference policy via a re-initialized LoRA proxy and a guided DPO loss mixed with SFT—and (ii) turning the improved captioner into a data generator to re-annotate videos and then SFT new models on these higher-quality captions. Caption quality is measured and rewarded with an atomic-event–based evaluation that decomposes captions into missing vs. hallucinated events. Models at 3B/7B achieve SOTA at comparable scales and the 72B model surpasses all open-source systems and rivals proprietary ones on benchmarks such as Video-MME, WorldSense, AVUT, Video-Holmes, DailyOmni, MLVU, and LVBench.

**Strengths:**

* **Principled, iterative preference learning for captioning.** MrDPO mitigates reference staleness by **merging the previous LoRA into the backbone and re-initializing a fresh LoRA proxy each round**, stabilized by a guided DPO loss with a small SFT term; ablations show cumulative reductions in caption error.
* **Task-aligned reward with atomic events.** The **missing + hallucination** metric operationalizes caption *completeness and factuality*, enabling LLM-judged preference pairs for RL.
* **Strong, broad empirical results and transfer.** The captioner beats strong baselines (incl. GPT-4o visual-only) on a human-validated caption metric and VDC-Detailed, and the **re-captioned data** lifts downstream QA across diverse benchmarks and scales.
* **Clear architecture for synchronized A/V modeling.** Interleaved audio/visual tokenization with frozen encoders and an aligner cleanly integrates audio without degrading visual performance.

**Weaknesses:**

* **Evaluator dependency and potential bias.** The atomic-event pipeline relies on **text LLMs** (e.g., GPT-3.5/4o) for both event extraction and preference decisions; while some human checks exist, the paper still **inherits evaluator bias/noise** and only partially audits it.
* **Caption→QA transfer hinges on data regeneration, not RL directly.** Authors note MrDPO mainly boosts captioning; general QA gains arrive **after** re-annotating and SFT, raising questions about how much MrDPO helps end-to-end QA absent the data-generation step.
* **Compute/engineering cost and robustness details.** Training uses substantial H800 resources across stages, and robustness to deployment artifacts (codec changes, audio dropouts) is not deeply probed.

**Questions:**

1. How convincing is the atomic-event LLM-evaluation pipeline as a reward signal—what additional **human studies, inter-evaluator agreement, or bias audits** would you require to fully trust the missing/hallucination estimates?
2. To what extent should the paper **disentangle** improvements from MrDPO itself versus the **re-captioned SFT data**? Which ablations (e.g., MrDPO-trained model evaluated on QA *without* regenerated data) would most clarify causal contribution?
3. Given the reported **H800 budgets**, what **efficiency analyses** (e.g., fewer rounds, smaller evaluators) and **robustness tests** (e.g., compression/noisy audio) would you consider necessary to judge real-world deployability?

---

> ### Author Response · Authors · 2025-11-21
> **Response to Reviewer 8DaU (part 1)**
>
> We appreciate the constructive feedbacks and would like to clarify each individual concerns raised by the reviewer as follows:
>
> > - Evaluator dependency and potential bias. The atomic-event pipeline relies on text LLMs (e.g., GPT-3.5/4o) for both event extraction and preference decisions; while some human checks exist, the paper still inherits evaluator bias/noise and only partially audits it.
> > - How convincing is the atomic-event LLM-evaluation pipeline as a reward signal—what additional human studies, inter-evaluator agreement, or bias audits would you require to fully trust the missing/hallucination estimates?
>
> We agree with the reviewer that all evaluation pipelines, human or model-based, inevitably contain some degree of noise. Our design goal is therefore not to eliminate noise entirely, but to use the most reliable and practical method currently available for long-form video caption evaluation. The tasks we study are primarily perception-level (watching and hearing), which reduces the impact of human subjective bias, and our atomic-event procedure follows a structured, checklist-style evaluation that further constrains evaluator variance. We provide several pieces of evidence supporting the effectiveness of this approach.
>
> 1. **Human alignment**: Appendix H reports a human Elo evaluation of captions. As shown in Table 11, the relative rankings produced by our atomic-event metric closely match human preferences, indicating strong alignment with human judgment.
> 2. **Cross-benchmark consistency**: The missing/hallucination estimates derived from our metric correlate well with results on established video understanding benchmarks such as VDC and VideoMME (Table 1(a)), suggesting that the metric captures meaningful aspects of video comprehension rather than evaluator-specific artefacts.
>
> Together, these results support the conclusion that the atomic-event–based evaluation is a reliable and interpretable reward signal for long-form video caption optimisation.
>
> > - Caption→QA transfer hinges on data regeneration, not RL directly. Authors note MrDPO mainly boosts captioning; general QA gains arrive after re-annotating and SFT, raising questions about how much MrDPO helps end-to-end QA absent the data-generation step.
> > - To what extent should the paper disentangle improvements from MrDPO itself versus the re-captioned SFT data? Which ablations (e.g., MrDPO-trained model evaluated on QA without regenerated data) would most clarify causal contribution?
>
> We list the video QA results of the SFT 7B model, the MrDPO-trained 7B model, the re-captioned SFT 7B model, the SFT 72B model and the re-captioned SFT 72B model of video-SALMONN 2+. Here, the regenerated captions are generated by the MrDPO-trained 7B model. Results are shown below:
>
> | Model               | VideoMME | AVUT     | WorldSense | DailyOmni | VideoHolmes | LVBench  | MLVU     |
> | ------------------- | -------- | -------- | ---------- | --------- | ----------- | -------- | -------- |
> | SFT (7B)            | 72.7     | **70.0** | 50.0       | 70.9      | 46.8        | 48.9     | 70.6     |
> | MrDPO (7B)          | 72.3     | 68.9     | 50.2       | 71.2      | **47.3**    | 49.7     | 70.9     |
> | ReCaption-SFT (7B)  | **73.4** | 69.5     | **50.9**   | **71.8**  | 46.9        | **49.7** | **73.6** |
> |                     |          |          |            |           |             |          |          |
> | SFT (72B)           | 78.8     | 71.8     | 55.4       | 77.4      | 56.0        | 55.4     | 77.6     |
> | ReCaption-SFT (72B) | **79.7** | **72.1** | **56.5**   | **79.4**  | **57.8**    | **55.5** | **80.4** |
>
> For 7B models, compared to the original SFT one, the QA performance of the MrDPO-trained model does not improve. This is because video QA likely focuses more on the model's reasoning abilities, whereas MrDPO is primarily used to optimize video captioning and does not train any further capabilities for reasoning. When re-captioning and training on the same 7B base model, we also observe no significant performance gain. This indicates that for a 7B LLM, due to the different capabilities required for captioning and QA, simply improving caption quality is not sufficient to significantly boost reasoning ability.
>
> However, for a more powerful LLM, such as a 72B model, its greater capacity allows it to naturally leverage the improvement in the quality of the caption training data to learn better reasoning skills, which in turn enhances its video QA performance.

---

> ### Author Response · Authors · 2025-11-21
> **Response to Reviewer 8DaU (part 2)**
>
> > - Compute/engineering cost and robustness details. Training uses substantial H800 resources across stages, and robustness to deployment artifacts (codec changes, audio dropouts) is not deeply probed.
> > - Given the reported H800 budgets, what efficiency analyses (e.g., fewer rounds, smaller evaluators) and robustness tests (e.g., compression/noisy audio) would you consider necessary to judge real-world deployability?
>
> **Regarding the efficiency:** First, the video-SALMONN 2 model will be open-sourced, so users will only need a small amount of computational resources for fine-tuning, without needing to train from scratch. If one wishes to reproduce the entire training pipeline while reducing computational costs, decreasing the number of MrDPO training rounds is a viable option. As shown in Table 3(a, b), the model's performance improves more significantly in the earlier rounds of MrDPO, with signs of convergence in the later rounds. Therefore, one could consider reducing the number of training rounds if achieving the absolute best metrics is not essential. As for the evaluator, since evaluating video captions based on atomic events is not a difficult task for existing LLMs, one could also consider replacing GPT-3.5 with a smaller open-source model, such as Qwen3-4B. Appendix C provides evaluation results using Qwen3-4B as the evaluator, showing its performance is close to that of GPT-3.5.
>
> **Regarding the robustness:** We believe the video-SALMONN 2 model has good robustness in complex scenarios, which primarily stems from its base model and training data. For instance, video-SALMONN 2+ is developed based on Qwen2.5-VL, and its audio encoder is based on the Whisper model. Since Qwen2.5-VL performs exceptionally well on the vast majority of visual tasks and Whisper has strong performance in speech recognition in noisy environments, the robustness of video-SALMONN 2+ in complex scenarios is well-assured. Furthermore, our video training set is based on existing video datasets covering various topics and domains, so the video-SALMONN 2 series models should also have no issues with real-world deployability. The excellent results on major video QA benchmarks serve as evidence of the robustness of the video SALMONN 2 series models.

---

> > ### Author Response · Authors · 2025-11-26
> >
> > Dear Reviewer 8DaU,
> >
> > We hope that our responses have adequately addressed the concerns you raised in your review, and we would greatly appreciate your further feedback. Please let us know if there is anything further we can clarify.
> >
> > Best regards, The Authors

---

### Official Review · Reviewer_N9PM · 2025-10-30

**Soundness:** 2
**Presentation:** 3
**Contribution:** 2
**Rating:** 4
**Confidence:** 5

**Summary:**

The authors propose MrDPO to enhance the model's video captioning capability, and leverage this capability to generate high-quality video caption data, which is then used to train a general-purpose model, thereby improving its general video question-answering ability.

**Strengths:**

1. video-SALMON 2 demonstrates strong video captioning capability, surpassing many well-known models such as Gemini 1.5, VideoLLaMA3, and Qwen2.5-VL.
2. video-SALMON 2 provides a self-constructed caption benchmark, which helps advance the video captioning capabilities of other models.
3. video-SALMON 2 exhibits powerful audio-visual question-answering performance, achieving strong results on VideoMME.

**Weaknesses:**

1. The paper’s first main contribution MrDPO (a DPO variant that updates the reference model during training to improve stability) is not novel; a similar approach was already proposed in TR-DPO [1]. Moreover, the paper’s primary focus is on enhancing video captioning performance, yet the design of MrDPO is unrelated to video captioning and appears to be a generic DPO method. The motivation behind MrDPO and the mechanism by which it improves video captioning capability remain unclear.

2. The second main contribution (using video captioning to enhance video question-answering performance) is also problematic. On one hand, this idea is already well established in the field and has been thoroughly validated in prior work. On the other hand, the authors fail to explain why or how improved video captioning capability leads to better video QA performance, leaving the underlying rationale unaddressed.

[1] Learn Your Reference Model for Real Good Alignment

**Questions:**

1. What is the design rationale behind the Video Caption Benchmark? Why did you develop a benchmark that evaluates video caption quality based on the extraction of various types of events? Is GPT-3.5 truly capable of reliably extracting diverse events from video captions?
2. Why does improving video captioning capability lead to enhanced video question-answering performance?

---

> ### Author Response · Authors · 2025-11-21
> **Response to Reviewer N9PM (part 1)**
>
> We appreciate the constructive feedbacks and would like to clarify each individual concerns raised by the reviewer as follows:
>
> > - The paper’s first main contribution MrDPO (a DPO variant that updates the reference model during training to improve stability) is not novel; a similar approach was already proposed in TR-DPO. Moreover, the paper’s primary focus is on enhancing video captioning performance, yet the design of MrDPO is unrelated to video captioning and appears to be a generic DPO method. The motivation behind MrDPO and the mechanism by which it improves video captioning capability remain unclear.
> > - Why does improving video captioning capability lead to enhanced video question-answering performance?
>
> First, we would like to emphasize that the main novelty of this paper lies in being the first to study high-quality long-form video caption generation. To support this, we introduce a new risk-based evaluation metric that measures caption completeness and hallucination with respect to atomic events, together with MrDPO and several additional algorithmic innovations designed specifically to optimise long-form video caption quality. These components form an integrated framework for both evaluating and improving caption generation at scale.
>
> Regarding the comparison to TR-DPO, the key difference is that TR-DPO updates the reference model but does not regenerate preference pairs, whereas MrDPO regenerates both the data and the preferences at each round, enabling the policy and reference models to co-evolve. Although MrDPO is in principle a generic DPO variant, in this paper we focus exclusively on validating its effectiveness for video captioning, and leave its application to other tasks for future work. The motivation for MrDPO stems from two considerations:
>
> 1. **Motivation for using DPO**: Prior work has demonstrated that DPO is an effective objective for sequence-level optimisation [1, 2].
> 2. **Motivation for the multi-round design**: In later stages of training, single-round DPO may cause the policy model to drift too far from the fixed reference model, violating the assumptions underlying DPO and causing early convergence. Multi-round DPO mitigates this by periodically updating the reference model and regenerating preference pairs, allowing continued optimisation.
>
> Finally, regarding why MrDPO improves video captioning capability, we attribute this largely to the reward formulation, namely our caption metric based on atomic events. Reinforcement-learning-style optimisation methods rely on a well-defined reward signal; once the reward reliably captures the desired behaviour (in our case, accurate and complete long-form video descriptions), the model can effectively learn to optimise toward it.
>
> **Regarding the answer to the second question:**
>
> Captioning and VQA share a substantial underlying requirement: accurate understanding and representation of video content. A model that produces precise, comprehensive descriptions naturally acquires stronger video comprehension, which directly benefits downstream video reasoning and question-answering tasks.
>
> [1] Zhang et. al. Direct Preference Optimization of Video Large Multimodal Models from Language Model Reward. arXiv 2404.01258
>
> [2] Pang et. al. Iterative reasoning preference optimization, In Proc. NeurIPS, 2024.

---

> > ### Author Response · Authors · 2025-11-21
> > **Response to Reviewer N9PM (part 2)**
> >
> > > The second main contribution (using video captioning to enhance video question-answering performance) is also problematic. On one hand, this idea is already well established in the field and has been thoroughly validated in prior work. On the other hand, the authors fail to explain why or how improved video captioning capability leads to better video QA performance, leaving the underlying rationale unaddressed.
> >
> > Our contribution in this part is to demonstrate that improving video QA performance can be achieved by intrinsically enhancing video understanding through long-form video caption optimisation within the same model. The full procedure is as follows:
> > 1. we first optimise the model’s video captioning capability;
> > 2. we then use this improved model to re-annotate the training set, generating higher-quality video captions;
> > 3. finally, we show that training on these improved captions in turn enhances the model’s video QA performance.
> >
> > This forms a self-iterative improvement loop that does not rely on any stronger external video LLMs.
> >
> > To the best of our knowledge, this self-iterative improvement loop, where **the model improves its own video understanding through caption optimisation and then uses its improved captions to further strengthen downstream reasoning**, has not been explored in prior work. Existing literature that leverages captioning to improve video QA typically depends on external SOTA video understanding LLMs (e.g., Gemini, GPT-4o) to re-label or augment training data. These methods are effectively a form of knowledge distillation from a superior model (like from human produced groundtruth labels), whereas our approach focuses on intrinsic capability enhancement without requiring a stronger teacher model. This distinction is fundamental to the novelty of our method.
> >
> > Regarding why improved video captioning leads to better video QA, the underlying rationale is that long-form captioning requires comprehensive, fine-grained understanding of video content. When the model becomes better at generating accurate, detailed, and low-hallucination captions, these improvements directly translate to higher-quality training data. In datasets such as LLaVA-Video-178k, whose captions were originally generated by GPT-4o, annotation errors and hallucinations are known to exist. Our caption-improved model produces descriptions with fewer errors and richer event coverage, effectively generating:
> > -  more accurate annotations, and
> > - a larger set of implicit QA pairs (each corrected or newly captured event corresponds to a QA-worthy fact).
> >
> > Training on this improved data naturally strengthens the model’s underlying video comprehension, which in turn leads to better video QA performance. This aligns with a well-established principle in deep learning: higher-quality supervision yields stronger downstream models.
> >
> > [1] Zhang et. al. Video Instruction Tuning with Synthetic Data, TMLR, 2025.

---

> ### Author Response · Authors · 2025-11-21
> **Response to Reviewer N9PM (part 3)**
>
> > What is the design rationale behind the Video Caption Benchmark? Why did you develop a benchmark that evaluates video caption quality based on the extraction of various types of events? Is GPT-3.5 truly capable of reliably extracting diverse events from video captions?
>
> Our goal is to evaluate whether the generated video captions accurately and comprehensively describe all events that occur in the video. The proposed atomic-event evaluation method begins with human annotators exhaustively listing the atomic events in each video. This list is then used as a reference during evaluation to assess the quality of the model’s generated caption. We adopt this approach for two main reasons:
>
> 1. Text-only evaluation: Assessing the presence or absence of atomic events is purely a text-based task, which current text LLMs can handle reliably.
> 2. Clearer metric design: This method transforms the challenging problem of evaluating long-form free-form text into a structured counting task over well-defined events, enabling a more interpretable and meaningful error-rate metric.
>
> In our experiments, GPT-3.5’s event-based evaluations show strong agreement with human judgments, indicating that GPT-3.5 can “reliably extract diverse events” from long captions within this setup. We further validate this in Appendix H through a Human Elo evaluation. As shown in Table 11, the Elo ratings are highly consistent with the total error rates produced by the GPT-3.5 atomic-event evaluation, providing additional evidence of its reliability.

---

> > ### Author Response · Authors · 2025-11-26
> >
> > Dear Reviewer N9PM,
> >
> > We hope that our responses have adequately addressed the concerns you raised in your review, and we would greatly appreciate your further feedback. Please let us know if there is anything further we can clarify.
> >
> > Best regards, The Authors

---

### Official Review · Reviewer_KarS · 2025-11-01

**Soundness:** 2
**Presentation:** 2
**Contribution:** 2
**Rating:** 2
**Confidence:** 4

**Summary:**

This paper introduces video-SALMONN 2, a audio-visual large language model (MLLMs) designed to improve video captioning and question answering. The contribution of this paper is Multi-round Direct Preference Optimization (MrDPO), a novel RL-based training strategy. This method iteratively refines the model's captioning ability by using a GPT-based reward signal with "atomic events" to measure caption completeness and factuality. Unlike standard DPO, which uses a fixed reference model, MrDPO periodically merges the latest model improvements (via a LoRA proxy) into the reference model, preventing "reference staleness" and enabling continual improvement. The authors then use their best MrDPO-trained model to distill its knowledge by re-annotating a large dataset, creating a high-quality caption corpus. This new corpus is used to SFT a final set of models (video-SALMONN 2+), which not only excel at captioning but also transfer these gains to achieve new state-of-the-art results on a wide range of video QA benchmarks.

**Strengths:**

- The paper is well-written and easy to follow.
- The proposed method achieves the best performance compared to other methods in diverse metrics and datasets. In particular, it shows better performance than GPT-4o.

**Weaknesses:**

- MrDPO seems to be very similar to the concept of GRPO. As GRPO updates and adopts old policy model, MrDPO uses the updated model as a reference model. Could you compare MrDPO with GRPO?
- I’m concerned about the complexity and cost of the MrDPO pipeline. The proposed MrDPO requires: evaluating the quality of every video caption using GPT-3.5. In comparison of using the verifiable reward function such as BLEU, ROUGE, the caption evaluation with GPT is time-consuming and cost-intensive.
- It would be better to include the prompt that evaluates the quality of video caption. Since finding out and evaluating atomic events in the video caption is a core component of the proposed algorithm, the prompt utilized for the validation should be included.
- Could the authors further clarify the distinction between the "gDPO" loss and the common practice of adding a standard SFT loss (using ground-truth data) as a regularizer to the DPO objective? The paper claims a distinction, but Equation 3 appears to be a standard DPO loss plus a standard SFT loss.

**Questions:**

Please refer to weaknesses section.

---

> ### Author Response · Authors · 2025-11-21
> **Response to Reviewer KarS**
>
> We appreciate the constructive feedbacks and would like to clarify each individual concerns raised by the reviewer as follows:
>
> > MrDPO seems to be very similar to the concept of GRPO. As GRPO updates and adopts old policy model, MrDPO uses the updated model as a reference model. Could you compare MrDPO with GRPO?
>
> First, we note that MrDPO was publicly released earlier than GRPO. While the two methods share some high-level similarities, their key difference lies in how data generation and evaluation are organised. MrDPO generates samples in batches and then performs quality assessment on the full batch, whereas GRPO evaluates sample quality within each training step, concurrently with sample generation.
>
> In training pipelines that rely on an external LLM for quality scoring, MrDPO can issue large-scale parallel requests to the evaluator, enabling significantly higher evaluation throughput. In contrast, GRPO is constrained by the limited number of samples generated per training step, reducing the degree of parallelism available for external evaluation and thus lowering overall efficiency.
>
> ---
>
> > I’m concerned about the complexity and cost of the MrDPO pipeline. The proposed MrDPO requires: evaluating the quality of every video caption using GPT-3.5. In comparison of using the verifiable reward function such as BLEU, ROUGE, the caption evaluation with GPT is time-consuming and cost-intensive.
>
> The goal of our work is to generate comprehensive and accurate long-form video captions, which are highly valuable for many downstream tasks. In this setting, the top-ranked text outputs produced by the LLM are typically very long (e.g., our model’s captions average over 500 words) and exhibit substantial flexibility in phrasing and structure. As a result, traditional n-gram–based metrics such as BLEU, ROUGE, and even CIDEr, which primarily assess surface-level sequence similarity or fluency, are not well suited for evaluating the correctness of atomic events described in such long, free-form text. For example, when evaluating our model on the MSR-VTT test set, both BLEU-4 and CIDEr scores fall below 1, despite the captions being accurate and detailed.
>
> To address this mismatch, we introduce an evaluation metric tailored to long-form video descriptions. By decomposing both the ground-truth and generated captions into lists of atomic events and comparing them using a strong LLM-based analyzer, the proposed metric provides a more reliable and fine-grained assessment of the semantic accuracy of the generated video descriptions. Moreover, evaluating captions based on atomic events using GPT-3.5 is not as costly as the reviewer implied. GPT-3.5 costs \$0.5 per 1M input tokens and \$1.5 per 1M output tokens. In our experiments, considering that each DPO iteration requires evaluating 13k sample pairs, with each sample containing approximately 2k input tokens and 2k output tokens, the total cost is (0.002 × 0.5 + 0.002 × 1.5) × 26,000 = \$104. This expense is significantly lower than GPU costs. Besides, data annotations for all multimodal LLMs require high costs. For example, LLaVA-Video uses large amounts of GPT-4o's API calls as well as human effort to annotate a total of 1.3M samples. In a sense, the annotation cost for video-SALMONN 2 is much cheaper.
>
> ---
>
> > It would be better to include the prompt that evaluates the quality of the video caption. Since finding out and evaluating atomic events in the video caption is a core component of the proposed algorithm, the prompt utilized for the validation should be included.
>
> The evaluation prompt is included in the link cited in the footnote at L377; specifically, it is available in the repository’s eval.py file.
>
> ---
>
> > Could the authors further clarify the distinction between the "gDPO" loss and the common practice of adding a standard SFT loss (using ground-truth data) as a regularizer to the DPO objective? The paper claims a distinction, but Equation 3 appears to be a standard DPO loss plus a standard SFT loss.
>
> As described in L276–L286, the proposed gDPO loss is indeed equivalent to adding an SFT loss on the ground-truth samples to the standard DPO objective. However, to the best of our knowledge, this formulation has not been systematically examined or formalised prior to this work. We are not aware of literature that studies this combined objective in depth; if such work exists, we would greatly appreciate pointers to it.
>
> In our paper, we explicitly define this objective as gDPO, analyse its behaviour, and provide a comprehensive empirical study in Section 5.2. We therefore regard the formalisation and investigation of this combined loss as an algorithmic contribution.

---

> > ### Author Response · Authors · 2025-11-26
> >
> > Dear Reviewer KarS,
> >
> > We hope that our responses have adequately addressed the concerns you raised in your review, and we would greatly appreciate your further feedback. Please let us know if there is anything further we can clarify.
> >
> > Best regards, The Authors

---

### Author Response · Authors · 2025-12-03
**Summary of Contributions and Rebuttal**

Dear Area Chair,

We sincerely thank all reviewers for their time and engagement with our work, and we are grateful to both the Area Chairs and Program Chairs for their careful oversight of the review process.

We regret, however, that several reviewers (KarS, N9PM, 8DaU) appear to have substantial misunderstandings about core aspects of our paper, leading to perplexing questions and unusually low scores. Although we provided clarifications early in the discussion period, these reviewers did not follow up before the discussion closed. Below, we summarise our contributions and clarify the major misunderstandings.

---
### Main Contributions

- **A new metric and optimisation method for video captioning.** We introduce an atomic-event–based caption accuracy/completeness metric and the MrDPO algorithm. Our metric correlates strongly with human judgments  (Appendix H, Table 11) and cross-benchmark evaluations, and MrDPO consistently improves detailed long-form video captioning.

- **Self-iterative captioning–to–QA improvement pipeline.** We present a novel self-iterative pipeline that first improves video captioning and subsequently enhances video QA performance.

- **Video-SALMONN 2: a family of audio-visual LLMs achieving state-of-the-art results.** Our 3B, 7B, and 72B models achieve SOTA performance on major audio-visual understanding benchmarks (Video-MME, WorldSense, AVUT, Video-Holmes, and DailyOmni) as well as silent-video benchmarks (MLVU and LVBench).


---
### Response to Major Misunderstandings from Reviewers

- **Reviewer KarS: Novelty of MrDPO and replaceability of our metric.** MrDPO was publicly released before GRPO and offers higher evaluation parallelism and more stable training. The reviewer's suggestion to replace our metric with BLEU/ROUGE reflects a fundamental misunderstanding: BLEU/ROUGE measure surface text similarity, which correlates poorly with the accuracy/completeness of long-video captions (with/ 500+ words on average generated by our model), rich in diverse, semantically flexible details. Our proposed metric evaluates the **semantic correctness of atomic video events**, which BLEU/ROUGE cannot capture.

- **Reviewer N9PM: Motivation and generalizability of MrDPO; misunderstanding of our QA contribution.** The motivation for MrDPO is clear: standard DPO is effective for long-text optimisation but limited in single-round settings. Our multi-round extension addresses these limitations, and extensive experiments show that multi-round DPO significantly outperforms single-round DPO. Generalizability to other tasks is beyond the scope of this paper, and we do not claim such universality. We only demonstrated effectiveness for video captioning.

    A more serious misunderstanding concerns our QA contribution. Reviewer N9PM characterised our approach as a well-known routine, yet contradictorily also questioned why improving the accuracy and completeness of captions would help QA. This contradiction itself indicates that the approach is **not** known or trivial. In contrast to prior work that distils from stronger multimodal LLMs, our method is explicitly **non-distilled**. Demonstrating that QA performance can be improved self-iteratively, without relying on any external teacher models, is a central and nontrivial contribution of our work.

- **Reviewer 8DaU: Use of GPT-3.5 for evaluation; relationship between RL and QA improvements.** Human evaluation and cross-benchmark results in the paper corroborate the reliability of GPT-3.5-based scoring. The concern that QA improvements do not stem "directly" from RL reflects a misunderstanding of capability transfer. Captioning and QA are distinct skills; improving captioning alone typically does not improve QA. Our method boosts QA by regenerating high-quality data from improved captions and retraining the model, thereby improving alignment between audiovisual tokens and textual representations. We also provide ablations in the rebuttal showing the contribution of MrDPO to QA without data regeneration.

- **Reviewer S8iw found no substantive issues**, and the minor questions have been fully addressed in the rebuttal.


---
Given the substantial misunderstandings in the initial reviews and the lack of subsequent discussion, we respectfully request the Area Chair to reassess our submission on the basis of its demonstrated contributions and our detailed rebuttal.

Thank you again for your consideration.

Best regards,
The Authors

---

### Meta-Review · Area_Chair_dr3f · 2026-01-06

**Summary:**

Reviewers questioned the novelty of MrDPO compared to existing variants like TR-DPO or GRPO and expressed concerns regarding the dependency on potentially biased LLM-evaluators. They also doubted whether captioning optimization directly translates to reasoning-heavy QA, citing the pipeline's high complexity and API costs.

**Reviewer Concerns:**

Addressed: The authors successfully clarified evaluation cost-efficiency, provided requested audio-modality ablations, and defined the gDPO loss structure.
Outstanding: The fundamental novelty of MrDPO over generic iterative refinement remains contested.

**Reviewer Scores:**

KarS (2) likely rises to 3 following cost and architectural clarifications. N9PM (4) and 8DaU (4) would likely maintain their scores as concerns over technical novelty and evaluator-dependency remain. S8iw (6) remains satisfied after the detailed breakdown of visual-only versus audio-visual performance.

---

### Decision · Program_Chairs · 2026-01-26

Reject